# Functional Ex Vivo Tissue-Based Chemotherapy Sensitivity Testing for Breast Cancer

**DOI:** 10.3390/cancers14051252

**Published:** 2022-02-28

**Authors:** Marjolijn M. Ladan, Titia G. Meijer, Nicole S. Verkaik, Zofia M. Komar, Carolien H. M. van Deurzen, Michael A. den Bakker, Roland Kanaar, Dik C. van Gent, Agnes Jager

**Affiliations:** 1Department of Molecular Genetics, Erasmus MC Cancer Institute, University Medical Center Rotterdam, 3000 CA Rotterdam, The Netherlands; m.ladan@erasmusmc.nl (M.M.L.); t.meijer@erasmusmc.nl (T.G.M.); n.verkaik@erasmusmc.nl (N.S.V.); z.komar@erasmusmc.nl (Z.M.K.); r.kanaar@erasmusmc.nl (R.K.); 2Oncode Institute, Erasmus University Medical Center, 3000 CA Rotterdam, The Netherlands; 3Department of Pathology, Erasmus MC Cancer Institute, 3000 CA Rotterdam, The Netherlands; c.h.m.vandeurzen@erasmusmc.nl; 4Department of Pathology, Maasstad Ziekenhuis, 3007 AC Rotterdam, The Netherlands; bakkerma@maasstadziekenhuis.nl; 5Department of Medical Oncology, Erasmus MC Cancer Institute, 3000 CA Rotterdam, The Netherlands; a.jager@erasmusmc.nl

**Keywords:** breast cancer, chemotherapy sensitivity testing, docetaxel, cisplatin, ex vivo, functional assay

## Abstract

**Simple Summary:**

Most breast cancer patients receive chemotherapy as part of their treatment. Unfortunately, treatment outcomes cannot be predicted with the current methods. Therefore, in the preset study, we explore the feasibility of a functional sensitivity test for the chemotherapeutic agents cisplatin and docetaxel on breast cancer tissue slices in culture. We show that these two agents need to be analyzed differently; cisplatin treatment resulted in cell death and a reduction in proliferation, whereas docetaxel could be assessed by determining the relative numbers of cells in mitosis. We also took the next step towards clinic application by adapting this test for biopsies from metastatic breast tumors. This test is now ready for a direct evaluation of its predictive value in clinical trials.

**Abstract:**

Background chemotherapy is part of most breast cancer (BC) treatment schedules. However, a substantial fraction of BC tumors does not respond to the treatment. Unfortunately, no standard biomarkers exist for response prediction. Therefore, we aim to develop ex vivo sensitivity assays for two types of commonly used cytostatics (i.e., platinum derivates and taxanes) on organotypic BC tissue slices. Methods: Ex vivo cisplatin sensitivity assays were established using organotypic tissue slices derived from the surgical resection material of 13 primary BCs and 20 fresh histological biopsies obtained from various metastatic sites. Furthermore, tissue slices of 10 primary BCs were used to establish a docetaxel ex vivo sensitivity assay. Results: Cisplatin sensitivity was assessed by tissue morphology, proliferation and apoptosis, while the relative increase in the mitotic index was discriminative for docetaxel sensitivity. Based on these read-outs, a scoring system was proposed to discriminate sensitive from resistant tumors for each cytostatic. We successful completed the cisplatin sensitivity assay on 12/16 (75%) biopsies as well. Conclusions: We developed an ex vivo cisplatin and docetaxel assay on BC slices. We also adapted the assay for biopsy-sized specimens as the next step towards the correlation of ex vivo test results and in vivo responses.

## 1. Introduction

Chemotherapy remains the cornerstone of breast cancer (BC) treatment, despite increasing possibilities for targeted therapies. However, in the (neo)adjuvant setting, overtreatment is a serious concern. The demand for predictive biomarkers is therefore high. Unfortunately, such biomarkers for classic chemotherapies do not yet exist despite extensive research [1], probably because of the multifactorial nature of their mechanisms of action.

As DNA sequence or gene expression analysis has not yet yielded validated biomarkers, the direct determination of tumor sensitivity using functional assays appears to be an attractive alternative. For this purpose, viable tumor material is cultured and treated ex vivo. For meaningful chemotherapy sensitivity measurements, the growth characteristics of the tumor should be preserved ex vivo. Furthermore, to be useful for the clinical practice, the turnaround time from obtaining a tumor sample to the final test result should be relatively short, i.e., preferably no more than two weeks. Therefore, we and others developed methods to culture organotypic tissue slices derived from BC specimens ex vivo for up to one week [2,3,4,5]. Our method maintained the breast tumor cells in their natural micro-environment and enabled ex vivo screening for chemotherapeutic drug sensitivities. As a first example, we developed a functional sensitivity assay for anthracycline-based chemotherapy on primary BC slices, which is currently being tested in a clinical proof-of-concept study (Trialnumber NL 5588) [2].

Taxanes and platinum salts are two other chemotherapeutic agents that are frequently used for BC treatment. Platinum salts, such as carboplatin and cisplatin, cause double-strand DNA damage (DNA interstrand crosslinks and breaks). Tumor cells with homologous recombination deficiency (HRD) are highly sensitive for this treatment as a result of their strongly reduced repair capacity. Deleterious mutations in the *BRCA1/2* and *PALB2* genes cause HRD, and could therefore be used for patient selection [6,7]. However, other reasons for an HRD phenotype exist, and it is not clear whether only HRD tumors are cisplatin sensitive [8,9]. Therefore, improved detection of platinum salt sensitivity using functional assays is dearly needed.

Taxanes have a different mode of action; they inhibit microtubule dynamics and mainly influence cell cycle progression. Static microtubules cannot mediate the segregation of chromosomes, causing a cell cycle block in mitosis [10]. Although some genes have been suggested to be involved in chemosensitivity, no molecular nor functional predictive tests have been developed yet [11].

Our aim is to develop ex vivo functional cisplatin and docetaxel sensitivity assays on BC tissue by determining the optimal read-outs for ex vivo response assessments and adapting the assays for use on biopsies of BC metastases. These advances set the stage for the clinical validation phase. 

## 2. Materials and Methods

### 2.1. Patient Derived Xenografts

Cisplatin-resistant and -sensitive patient-derived xenograft (PDX) models (T250) were established at the Netherlands Cancer Institute, as previously described [12]. 

### 2.2. Primary Breast Cancer (BC) Specimens

Residual primary BC tissue was prospectively collected from wide local excision or ablation of the breast in the Erasmus MC Cancer Institute and Maasstad Hospital in Rotterdam, The Netherlands. After the macroscopic evaluation of the surgical specimen by pathologists, fresh residual tumor tissue was collected for research purposes, according to the “Code of proper secondary use of human tissue in the Netherlands” established by the Dutch Federation of Medical Scientific Societies. This was approved by the local Medical Ethical commission of the Erasmus MC (MEC-11-098). Patients who had objected to the secondary use of residual tumor material for research purposes were not included in this study. 

### 2.3. Metastatic Breast Cancer Biopsies

Patients with recurrent or metastatic BC who were planned to start systemic treatment and who had metastatic lesions amenable for biopsies were eligible for the HRD study (Dutch Trial Register number: NTR5574 [13]) or the RECAP study (Dutch Trial Register number: NTR6560). Both studies (NL49306.078.14/MEC14-295 and NL60293.078.17/MEC17-213) were approved by the local Medical Ethical Commission. After written informed consent and registration, each patient was scheduled for a biopsy of a metastatic lesion. If a second biopsy could be obtained, this was used for this study for drug sensitivity screening on organotypic tissue slices. The biopsy was performed by a (intervention) radiologist according to local protocols. For distant metastases, a core needle biopsy with a minimum of 18 gauge and a maximum of 14 gauge was performed under imaging guidance. Biopsies from superficial (skin and subcutaneous) BC recurrences were also allowed and performed using a standard 4 mm biopsy puncher. 

### 2.4. Tissue Slicing and Drug Treatment Ex Vivo

Tumor samples were collected in customized breast medium, after which slices were generated using a Leica VT 1200S Vibratome as described previously [2]. Metastatic BC biopsies were embedded in 4% low melting agarose in PBS at 37 °C under a shallow angle before slicing. Slicing resulted in 3–5 slices from a biopsy (300 µM thick and several mm in length). Culturing was performed at 37 °C in a 5% CO_2_ atmosphere under constant rotation at 60 rpm using a Stuart SSM1 mini orbital shaker. 

For the development of the drug sensitivity assay, tissue slices were cultured for three days with a constant concentration of cisplatin (Accord Healthcare, Utrecht, The Netherlands) or docetaxel (Biovision, ITK Diagnostics, Uithoorn, The Netherlands). Proliferating cells for all conditions were labeled by adding 30 μM 5-Ethynyl deoxyuridine (EdU) (Invitrogen, Carlsbad, CA, USA) to the culture media during the last 2 h, before fixation for cisplatin treatment and during the last hour before fixation for docetaxel treatment. Tumor slices were fixed in 10% neutral buffered formalin for at least 24 h at room temperature. Subsequently, tumor slices were embedded in paraffin and 4 μm sections were generated.

### 2.5. Staining Protocols

Histological tumor architecture was examined by hematoxylin and eosin (HE) staining. Proliferating tumor cells were visualized by immunofluorescent staining with anti-PanCytokeratin (AE1/AE3) (Santa Cruz Biotechnology, Dallas, TX, USA, sc-81714, diluted 1/500) and Goat anti-Mouse Alexa Fluor 488 (Thermo scientific, 1/1000 dilution) as a secondary antibody, and a chemical staining protocol (Click-it staining) to visualize EdU incorporation, as described previously [2]. Apoptosis was visualized using a terminal deoxynucleotidyl transferase dUTP nick end-labeling (TUNEL) assay, as described previously [2]. Phospho-H3 (p-H3) immunostaining was performed using an anti-phospho serine10 histone H3 antibody (Millipore, 1/500 dilution) and Goat anti-Rabbit Alexa Fluor 488 (Thermo scienctific, 1/1000 dilution) as a secondary antibody. This staining was combined with the click-it reaction with Atto 594 (Invitrogen, manufacturers protocol) to detect EdU-positive cells. 

### 2.6. Image Acquisition and Analysis

HE stainings were imaged by light microscopy. For the analysis of HE staining, the whole tissue slice was evaluated as described previously [2]. 

For EdU and TUNEL staining quantifications, 3–12 random images (200× magnification for larger tumor specimens and 400× magnification for biopsies) from each tumor slice were generated for quantification purposes using an Imager D2 wide field near-infrared microscope (Zeiss, Oberkochen, Germany). Levels of apoptosis, expressed as TUNEL positive DAPI pixels, were quantified by analysis of TUNEL microscopy images as previously described [14]. The ratio of keratin positive cells that were also EdU positive were quantified manually using the counting tool in Adobe Photoshop CC v19.0 (Adobe Inc., San Jose, CA, USA).

For p-H3/EdU staining, random fields of view were photographed and the relative numbers of EdU- positive and p-H3-positive nuclei were quantified manually using the counting tool in Adobe Photoshop CC v19.0 (Adobe Inc., San Jose, CA, USA).

### 2.7. Defining Ex-Vivo Sensitivity Test Results

Ex vivo sensitivity scores were determined for HE, EdU and TUNEL staining. Various cut-off values were explored and the overall cisplatin sensitivity score was calculated as the mean of the three parameters for a chosen cut-off. The outcome measures received a number: sensitive = 1, intermediate = 2 and resistant = 3. The mean was then calculated for each sample and rounded to the nearest number. For example, if a sample was HE sensitive (1), EdU sensitive (1), but TUNEL resistant (3), the mean was 1.67, and therefore the sample was scored as intermediate.

### 2.8. Statistics

Statistical analyses were all 2-sided and performed using SPSS statistics v21.0 (IBM, Armonk, NY, USA) or Graphpad Prism v6.0 (Graphpad Software Inc., San Diego, CA, USA) for analyzing the differences in the graphs. Significance was calculated by Fisher’s exact test for categorical data and by the Mann–Whitney test for continuous data. The *p*-values of <0.05 were considered significant.

## 3. Results

### 3.1. Ex Vivo Cisplatin Drug Screening on Tissue Slices Reflects an In Vivo Cisplatin Response in PDXs

We previously measured the anthracycline sensitivity of primary BC samples based on tissue morphology, proliferation and apoptosis after the treatment of freshly cut slices ex vivo [2]. We first adapted this assay for cisplatin, the most potent platinum salt used for anti-tumor treatment. 

We first established the optimal conditions for the ex vivo sensitivity assay in organotypic tissue slices from PDX tumors with known in vivo cisplatin sensitivity. Organotypic tissue slices from a cisplatin-sensitive and a resistant PDX tumor were exposed to various concentrations of cisplatin for three days. The tissue response to cisplatin treatment was determined by the analysis of EdU incorporation (proliferation), TUNEL (apoptosis) and HE staining (morphology). The morphology of the tumor tissue was assessed by scoring aberrant nuclear morphology after HE staining and the tissues were categorized as ‘intact’ or ‘deteriorated’ for each treatment condition. Proliferation was measured by EdU pulse labeling at the end of the incubation period, which is a measure for cells in the S phase of the cell cycle after three days of treatment (Appendix A). We examined which concentrations of cisplatin discriminate best between sensitive and resistant tumors after three days. Figure 1A,B shows that the responses of the in vivo resistant and sensitive PDX tumor to ex vivo cisplatin treatment significantly differ, indicating that ex vivo cisplatin drug screening on PDX tumor slices accurately reflects their in vivo drug response. The 10 μg/mL cisplatin concentration resulted in tissue deterioration (HE staining) in both tumors and very low proliferation in both sensitive and resistant tumor (Appendix A). Therefore, this concentration was not considered useful for discriminating sensitive and resistant tumors. As the effect of cisplatin on proliferation already occurs at a lowest concentration and the effect on apoptosis becomes more apparent at higher concentrations of cisplatin, the 1 μg/mL and 5 μg/mL concentrations were selected for the ex vivo assay on human tumor samples. Only if sufficient amounts of material were available, other concentrations were tested as well.

### 3.2. Cisplatin Sensitivity on Surgical Resection Material

Tissue reactions were assessed for surgical samples of 13 primary BC tumors after 3 days of treatment, with cisplatin concentrations ranging from 1 to 10 µg/mL.

The concentration at which the change between ‘intact’ and ‘deteriorated’ occurred on HE staining was generally clear and relatively uniform throughout the tumor tissue, and in most cases either occurred at 5 µg/mL or it was not even visible at 10 µg/mL cisplatin (Figure 2A). 

As tumors are highly heterogeneous in proliferation rate, the proliferation rates between tumors could not be compared directly. Therefore, we used the percentage EdU-positive cells in the untreated slice as the reference value and explored how tumors would be classified when different percentages of reduction in the proliferation rate compared to the untreated sample were classified as ‘significant reduction’ (Appendix A). A 70% reduction resulted mainly in samples that reached this point only at the highest concentration. There was not much difference between cut-off values of 50 and 60%, while a 40% reduction resulted in a relatively large number of samples that would already reach a significant reduction at the lowest cisplatin concentration and only a few samples classified as resistant. Therefore, we decided to use a 50% decrease in the EdU signal for this first analysis (Figure 2B). 

Although apoptosis is an important parameter for response assessment, it is inherent to the set-up of the ex vivo tissue sensitivity assay that all tissue slices, including the untreated slices, show heterogeneous levels of apoptosis, even without chemotherapy treatment. In a previous cohort of tumor samples as well as in this cohort, untreated samples showed up to 40% TUNEL-positive cells (Appendix A) [2]. We tried to take this into account using two different scoring methods. First, we analyzed the data by using fixed thresholds between 20 and 60% TUNEL-positive nuclei before scoring the sample as apoptotic (Figure 2C and Appendix AA–E). As a second method, we scored an increase in at least 10–50% relative to the untreated sample as apoptotic (Figure 2D and Appendix AF–J). With only the data of the primary BC slices, it is difficult to choose between both methods. For this article, we decided to use the ≥20% increase parameter for the subsequent analyses. 

The next question was how to define the cut-off values for the above-mentioned parameters to discriminate sensitive, resistant and intermediate tumors. Determining the cut-off value for the cisplatin chemotherapy sensitivity assay was relatively difficult, as the differences between tumors were not very pronounced. A distinction could be made between samples that were clearly deteriorated at 5 µg/mL and samples that still did not show a significant effect at this concentration (and mostly also at 10 µg/mL). Therefore, we decided to classify tumors that did not yet reach a significant effect for the parameter at 5 µg/mL as ‘resistant’. Some tumors already showed significant effects at 1 µg/mL, which were therefore classified as ‘sensitive’. Other samples were tentatively classified as ‘intermediate’ (Figure 3A). Taking the three parameters (morphology, proliferation and apoptosis) together, resulted in an average ex vivo score. When one of the three parameters could not be scored, the test outcome was still considered valid if only one was missing and the remaining two parameters were concordant (Figure 3B and Appendix A).

### 3.3. Docetaxel Sensitivity Assay 

Tumor slices of primary BC samples from 10 different patients were obtained and incubated with increasing concentrations of docetaxel (1–1000 nM) to determine differences in docetaxel sensitivity. Unexpectedly, we did not observe an induction of apoptosis or decrease in proliferation at 3 or even 8 days of treatment (Appendix A). Subsequently, we reasoned that the inhibition of microtubule dynamics by taxane treatment should cause an inability to complete mitosis. Mitotic cells were visualized by immunofluorescent staining for phosphorylated histone H3 (p-H3), a mitosis specific histone modification. Although differences in relative numbers of p-H3-positive cells were observed, this analysis was hampered by vastly divergent proliferation rates between the tumor samples. Therefore, we combined the p-H3 staining with EdU staining and corrected for the initial proliferation rate by taking the ratio of the EdU-positive and p-H3-positive cells (Figure 4). Some tumors showed a decrease in the EdU/p-H3 ratio at very low concentrations of docetaxel (Figure 4B left panel), while other tumors showed this effect only at the higher concentrations (Figure 4B right panel). This change was caused by the increase in mitotic cells, while the percentage of EdU-positive cells did not decrease significantly over the course of the treatment. The best discrimination between samples was observed at 10 nM docetaxel (Figure 4C), where we observed the clearest differences in the response between the various tumor samples. At 1 nM docetaxel, the response was more heterogeneous, while 100 nM docetaxel caused a dramatic decrease in this ratio for all the samples.

Clinical decision making would require the determination of a cut-off value for resistant versus sensitive tumors. The differences in docetaxel sensitivity were rather high. A cluster of samples showed a drop to less than 30% of the untreated ratio at 10 nM docetaxel. This number is close to what one would expect for the fraction of highly sensitive tumors (Figure 4C) [15]. Two tumors remained above 80% of the initial ratio, suggesting that they are the intrinsically resistant tumors. The group of four tumors between these groups was tentatively categorized as intermediate (Figure 4D, Appendix A). A definitive choice for the cut-off values requires a correlation with the clinical response. 

### 3.4. Sensitivity Assays on Biopsies

As a first step towards clinical studies, the adaption of the sensitivity assays for use on very limited tissue acquired by core needle biopsies from metastatic BC lesions was needed. We first developed a method to obtain 3–5 slices from a single 14–18 gauge (G) needle biopsy in order to incubate the tissue with different concentrations of chemotherapeutics (Figure 5A). To achieve this technically demanding task, the biopsies were embedded in agarose under a shallow angle, and subsequently 300 μm tissue slices were generated. At the moment the biopsy assay was developed, we did not yet have a sufficiently developed taxane sensitivity assay. Therefore, slices were incubated for three days without cisplatin or with 1 or 5 µg/mL cisplatin. After treatment, tissue morphology, proliferation and apoptosis were analyzed, as described for the primary BC specimens.

Tissue slices were generated from 20 metastatic core needle biopsies. Four biopsies contained insufficient or no tumor cells (*n* = 4) to perform the test. In the 16 biopsies containing sufficient numbers of tumor cells, 12 ex vivo tests were successful (75%; Figure 5B), 2 tests were partly successful (not all the conditions contained tumor cells, generating test results for the untreated and 1 μg/mL cisplatin condition only) and 2 biopsies were not successful because the degree of necrosis was too high in the untreated sample. Tumor biopsies were derived from different metastatic sites, including the liver, lymph node and chest wall (Appendix A). Successful tests were obtained from any of these locations. All unsuccessful tests were obtained from 18G needle biopsies; all biopsies obtained with 14G needles were successful compared to only 50% of the biopsies obtained with 18G needles (Appendix A). 

Next, we used the same criteria as mentioned in Figure 3A for the cisplatin ex vivo sensitivity test that were established in primary tumors. The morphology, proliferation and apoptosis showed a similar range of sensitivities in tumor biopsies, although they were in general slightly more sensitive. 

Of the 12 successful ex vivo cisplatin sensitivity tests, four biopsies were scored as ex vivo cisplatin resistant, six intermediate and two sensitive (Figure 5C and Appendix A). In addition, there were two biopsies in which not all the tissue slices contained tumor cells. From the two partly successful test results, we determined that these tumors were not sensitive. The definitive cut-off values for the cisplatin sensitivity assay should be re-evaluated in clinical validation studies. 

## 4. Discussion

We developed a functional ex vivo sensitivity test for cisplatin and docetaxel using tumor slices derived from primary surgical resection material and metastatic biopsies. We observed large differences in the ex vivo sensitivity, leading us to propose a read-out system that combines morphology, proliferation and apoptosis induction for cisplatin, while docetaxel effects can be measured by determining the relative numbers of cells in mitosis and the S phase of the cell cycle. This highlights the need to take into account the mechanism of action of a particular chemotherapeutic drug to set up functional ex vivo sensitivity tests. 

We observed differences in cisplatin sensitivity between the BC slices, mainly in the 1 and 5 µg/mL concentration range. Therefore, we decided to develop a preliminary scoring system in which these two concentrations were used to classify tumors as ex vivo sensitive, intermediate or sensitive. The precise cut-off values for proliferation and apoptosis are still somewhat arbitrary. Based on the first study, we decided to use a reduction in EdU-positive nuclei by at least 50% and an increase in TUNEL-positive nuclei of at least 20% of the total tumor cell population as a starting point for further studies. However, we still consider the option to use a fixed cut-off of 40% TUNEL-positive nuclei. With the current knowledge, it is difficult to choose between these criteria, due to the small number of samples studied and the lack of clinical data. In the current cohort, both scoring methods would not change much in the classification of tumor sensitivity. Ultimately, clinical validation is required to pinpoint the exact effective cut-off values. 

To date, ex vivo chemotherapy sensitivity assays were performed on larger pieces of tissue from surgical resection material. It is of utmost importance that an assay can be performed on minimal amounts of material, such as core needle biopsies, to be of any clinical interest. Furthermore, ex vivo sensitivity results should be obtained in less than two weeks to allow the implementation in the clinical decision making process. The current study shows that chemotherapy sensitivity assays are possible on biopsies from metastatic BC lesions within this time frame. The success rate of the assay can most probably be improved considerably by using a larger needle size (14G) and/or taking multiple biopsies. These technical improvements are a major step towards clinical proof of concept studies in the neoadjuvant or metastatic setting. 

Tumor heterogeneity or differences between individual metastatic lesions in one patient may hamper the correlation between ex vivo sensitivity and in vivo response of the tumor. Some of this heterogeneity could be observed in slices of the primary tumor. This was taken into account by selecting random areas throughout the tumor slice for image analysis. Needle biopsies contain far less tissue and therefore they show less heterogeneity. Thus heterogeneity cannot be assessed by this sensitivity assay (or any other assay that depends on analysis of biopsies). Future clinical studies have to clarify whether this is a major factor influencing the development of a predictive chemotherapy sensitivity assay. On the other hand, our ex vivo characterization of tumor samples might help to correlate tumor sensitivity to gene expression or other genomic data. It would separate tumor-intrinsic data from systemic factors (such as the metabolic processing of drugs in the liver) that might otherwise hamper the interpretation of genomic data derived from tumor samples.

Ex vivo sensitivity assays are widely applicable in both translational research and personalized medicine. We included cisplatin and docetaxel treatments in the current study and anthracyclines in an earlier study [2], and there is no reason to suspect that other (chemotherapy) treatments would pose major problems. A previous study stated that taxanes might be hard to study ex vivo because they inhibit cell growth through a different mechanism of action than DNA damaging chemotherapy [16], which limits the value of measuring proliferation and apoptosis. In the present paper, we show that this problem can be solved by measuring cell cycle block in mitosis instead of cell death or proliferation parameters. Ex vivo sensitivity assays can be used for various tumor types, such as prostate tumors [14], lung tumors [17,18] and head-and-neck tumors [19,20,21]. Culture media and precise incubation conditions should be optimized for each tumor type and treatment. However, we did not yet encounter tumors that cannot be cultured for several days to weeks ex vivo. They retain in vivo tumor complexity, including tumor heterogeneity and the original microenvironment [22], also making this culture system attractive for fundamental cancer research. 

There are several limitations in the approach presented in this study. First, the already-mentioned needle size, which must be at least 14G to obtain sufficient material for the ex vivo test. Furthermore, the quality of the tumor material itself is important (e.g., too much tumor necrosis in the untreated tumor slices). Other limitations of the organotypic tissue slice system are the fact that pharmacokinetic conditions in vivo are difficult to mimic ex vivo and that it is a low-throughput technique; a limited number of slices can be obtained from one core needle biopsy. In the future, the latter issue can be overcome by advances in live-imaging microscopy, a technique that allows a single tissue slice to be analyzed at multiple time points. Moreover, an adaptation to the cancer-on-chip approach could eventually reduce the amount of tumor material required for the analysis of tumor sensitivity, as well as create a more controllable environment to mimic in vivo pharmacokinetic conditions [22,23].

## 5. Conclusions

In conclusion, we developed a functional ex vivo sensitivity test for cisplatin and docetaxel using tumor tissue slices of BC patients. The current study also shows that chemotherapy sensitivity assays are technically possible on limited amounts of tissue in biopsies from metastatic BC lesions. Rather than predicting a response based on certain biomarkers, the ex vivo sensitivity assays directly measure the response of individual tumors to a certain therapy. To develop this method further, clinical trials should include a biopsy for organotypic tissue slices to further optimize tissue cultivation methods and scoring systems. In line with this, we recently conducted a clinical proof-of-concept trial (Available online: Trialregister.nl/trial/5588 (accessed on 19 January 2022)), specifically powered to determine the predictive value of the ex vivo anthracycline sensitivity assay that uses organotypic slices from biopsies. In the future, these tissue-based ex vivo sensitivity assays have the potential to individualize therapy response prediction for patients in clinical practice and can be extended to other cancer types and treatments. 

## Figures and Tables

**Figure 1 cancers-14-01252-f001:**
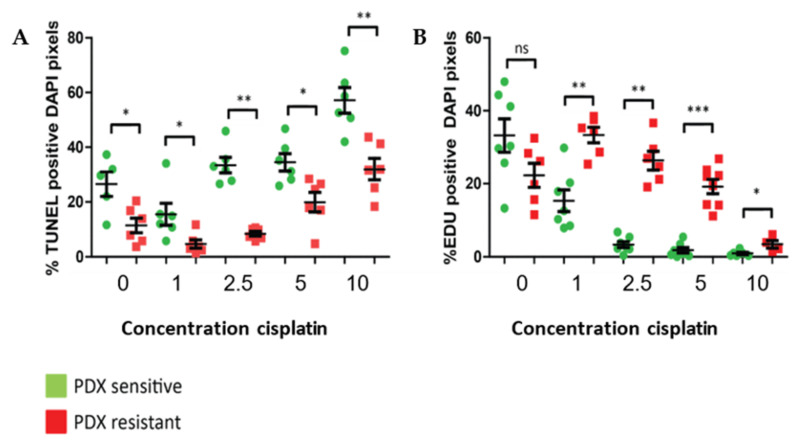
In vivo sensitive and resistant PDX tumors show differential responses to ex vivo cisplatin treatment. (**A**) EdU-positive cells in the tissue slices. (**B**) Quantification of the fraction of respective TUNEL-positive DAPI pixels. Six image fields were analyzed per tumor slice. The graphs show each point (representing one image field) with the mean and SEM. * *p* < 0.05, ** *p* < 0.01, *** *p* < 0.001, ns = non-significant.

**Figure 2 cancers-14-01252-f002:**
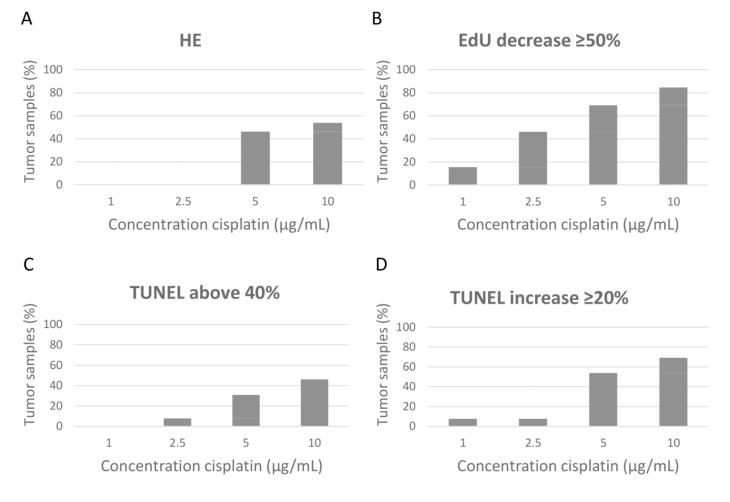
Cisplatin-sensitivity scoring of primary breast cancer samples (*n* = 13). (**A**) Morphology based on hematoxylin and eosin (HE) staining, scored as mostly deteriorated. (**B**) Proliferation based on EdU incorporation, determining the lowest cisplatin concentration, which reached a decline of ≥50% EdU-positive cells compared to the untreated sample. (**C**,**D**) Apoptosis based on TUNEL staining, determining the lowest cisplatin concentration, where TUNEL positivity reached ≥40% TUNEL-positive DAPI pixels (**C**), or the lowest cisplatin concentration where TUNEL positivity reached ≥20% increase relative to the untreated sample (**D**). Tumor samples (%) scores the cumulative number of tumor samples that reached the threshold at that concentration.

**Figure 3 cancers-14-01252-f003:**
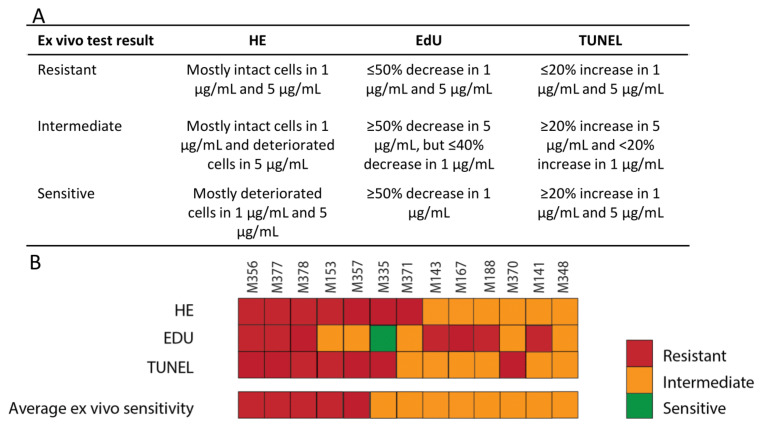
Overall cisplatin sensitivity scores of the primary breast cancer samples. (**A**) Criteria for ex vivo cisplatin sensitivity scoring. (**B**) The HE, EdU, TUNEL and average ex vivo sensitivity scores using the criteria defined in (**A**). M-numbers represent individual primary mammary tumors.

**Figure 4 cancers-14-01252-f004:**
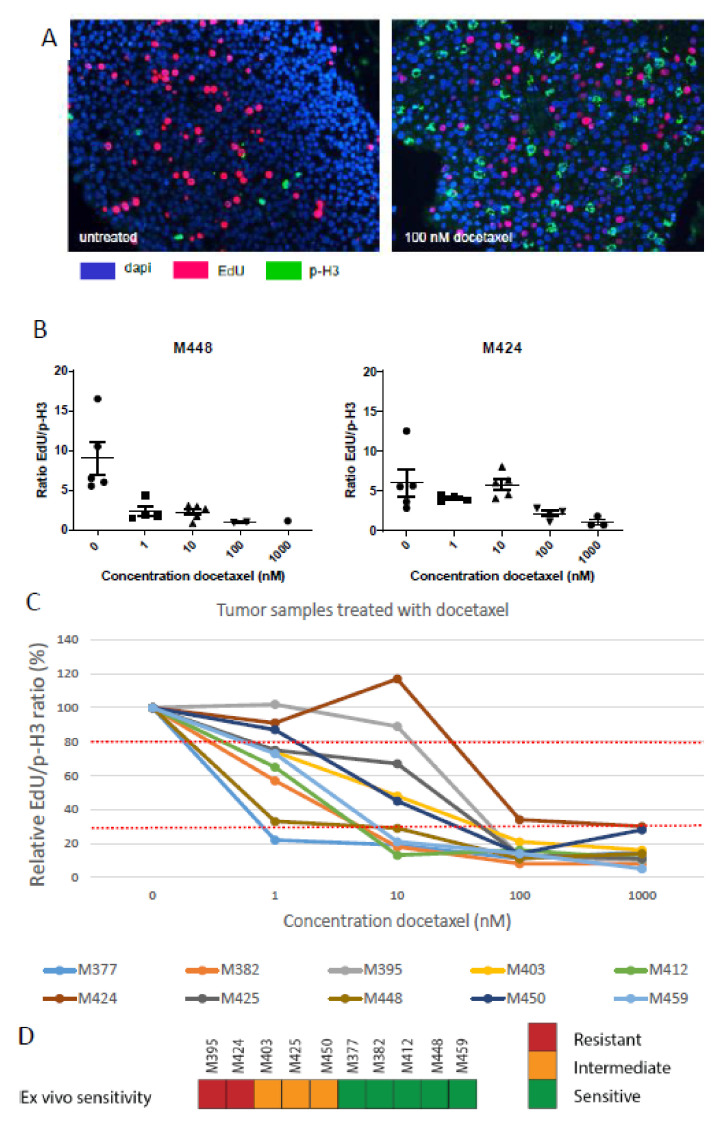
Ex vivo docetaxel treatment of primary breast cancer. (**A**) Typical microscopic image of DAPI, EdU and p-H3 staining of primary BC slices without and with 3 days of 100 nM docetaxel treatment. (**B**) EdU/p-H3 ratios of two primary BC samples incubated for three days with various docetaxel concentrations. Each data point (circle, triangle or square) is the score for one microscopic field of view, with the mean and SEM indicated for each docetaxel concentration. (**C**) The EdU/p-H3 ratio in response to 3 days of incubation with the indicated docetaxel concentrations relative to the untreated control for 10 primary BC samples. Dotted red lines indicate the proposed thresholds for optimal discrimination between sensitive, intermediate and resistant tumors. (**D**) Overall results of ex vivo docetaxel sensitivity in the primary tumors using 30% and 80% of the relative EdU/p-H3 ratio at 10 nM docetaxel as thresholds for discriminating sensitive, intermediate and resistant tumors. M-numbers represent the individual primary mammary tumors.

**Figure 5 cancers-14-01252-f005:**
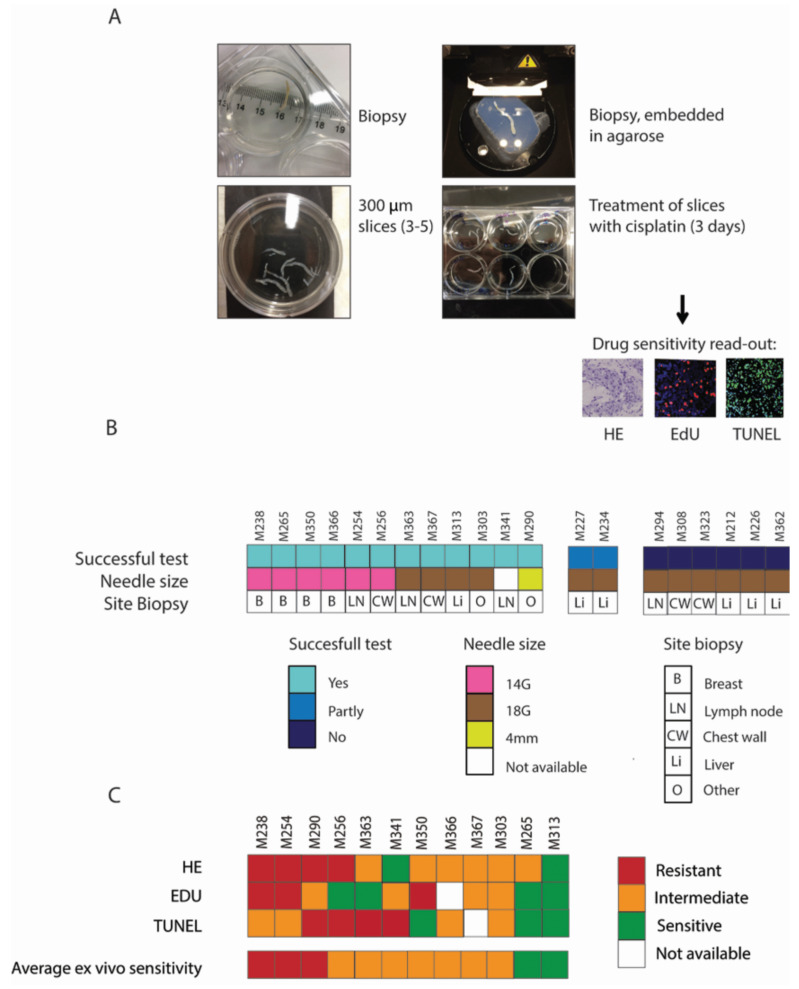
Feasibility of the ex vivo sensitivity test on organotypic tissue slices from biopsies. (**A**) Workflow of the ex vivo drug sensitivity screening. Biopsies were embedded in agarose, positioned horizontally, and 300 µM tissue slices were generated. Tissue slices were treated with cisplatin for 3 days and EdU was added 2 hours before fixation. Subsequently, the tissue was formalin fixed and paraffin embedded (FFPE). Drug sensitivity read-out consisted of HE, EdU and TUNEL stainings. (**B**) Success of the test in relation to the needle size used for the biopsy and the tumor site where the biopsy was taken. A successful test was achieved when sufficient numbers of tumor cells were present in the untreated, 1 μg/mL and 5 μg/mL cisplatin-treated tissue slices. When not all the conditions contained tumor cells, but only the untreated and 1 μg/mL cisplatin condition, the test was considered to be partly successful. When a biopsy contained very little or no tumor cells (*n* = 4) or was necrotic (*n* = 2), the test was not successful. (**C**) Outcome per separate test (HE, EdU and TUNEL) and overall test results, based on the criteria defined in Figure 2A, for all samples with a successful test. M-numbers represent individual mammary tumor biopsies.

## Data Availability

All data can be found in the manuscript.

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
