# Peer review of "Functional Ex Vivo Tissue-Based Chemotherapy Sensitivity Testing for Breast Cancer"

_cancers, 2022, doi:10.3390/cancers14051252_

Round 1
Reviewer 1 Report
The proof-of-concept study by Marjolijn M. Ladan et al. demonstrates the feasibility of ex vivo breast cancer cell sensitivity to cisplatin and docetaxel. The authors obtained a range of drug sensitivities in a timely manner amenable to testing in the clinical setting. Finally, the authors correctly identify the missing clinical validation of their results as a limitation to their study.
I only have minor recommendations, mainly for discussion purposes:
- The x-axis of figure 1 should be re-labelled to avoid the confusing duplication of cisplatin doses.
- The sentence at lines 406-408 requires a grammatical check: "Moreover, adaptation to a cancer-on-chip approach could eventually the amount of tumor material required for analysis of tumor sensitivity, as well as create a more controllable environment to mimic in vivo pharmacokinetic conditions [22,23]."
- The study highlights the difficulty in setting meaningful cut-off values for tissue integrity, arrest, proliferation and apoptosis. The authors argue that increasing the number of samples and clinical validation will be important to fine-tune these cutoffs. Alternatively, have the authors considered statistical methods to determine cut-off values such as ROC curve analysis ?
- The authors have generated ex vivo breast cancer specimens with known sensitivities to cisplatin and docetaxe. Thus, material is available to determine a gene signature of drug sensitivity. In the future, one may consider a drug sensitivity score based on a combined gene signature score + ex vivo sensitivity score. Please discuss.
Author Response
We thank the review for the constructive criticism to our paper and the positive evaluation of our study. Our detailed answers are as follows:
- The x-axis of figure 1 should be re-labelled to avoid the confusing duplication of cisplatin doses.
The figure has been relabeled.
- The sentence at lines 406-408 requires a grammatical check: "Moreover, adaptation to a cancer-on-chip approach could eventually the amount of tumor material required for analysis of tumor sensitivity, as well as create a more controllable environment to mimic in vivo pharmacokinetic conditions [22,23]."
The word ‘reduce’ was added (could eventually reduce the amount). Thanks for pointing this out.
- The study highlights the difficulty in setting meaningful cut-off values for tissue integrity, arrest, proliferation and apoptosis. The authors argue that increasing the number of samples and clinical validation will be important to fine-tune these cutoffs. Alternatively, have the authors considered statistical methods to determine cut-off values such as ROC curve analysis ?
We think that statistical methods would still need clinical data for comparison. Otherwise, there is no validation to real life data and only confidence intervals for the technical variability can be set, which do not necessarily help to find cutoffs for the predictive test.
- The authors have generated ex vivo breast cancer specimens with known sensitivities to cisplatin and docetaxe. Thus, material is available to determine a gene signature of drug sensitivity. In the future, one may consider a drug sensitivity score based on a combined gene signature score + ex vivo sensitivity score. Please discuss.
This is indeed an interesting point. We added a short discussion of this point (which would merit more elaborate investigation in the follow-up):
On the other hand, our ex vivo characterization of tumor samples might help to correlate tumor sensitivity to gene expression or other genomic data. It would separate tumor-intrinsic data from systemic factors (such as metabolic processing of drugs in the liver) that might otherwise hamper interpretation of genomic data derived from tumor samples. (lines 389-393)
Reviewer 2 Report
General comments:
The authors explored the feasibility of a functional sensitivity test for the chemotherapeutic agents cisplatin and docetaxel on breast cancer tissue slices in culture.
Major comments:
- The overall cisplatin sensitivity score is not clear. Please provide a detailed information for it in section of Materials and methods.
- Is these tissue-based ex vivo sensitivity assays only suitable for breast tumor? Please also discuss about the potential application to other cancers.
- line304: What are the detailed steps for embedding biopsies in agarose?
- Do the cells alive in the tissue-based ex vivo sensitivity assays? Because the authors use tissue-based ex vivo sensitivity assays to treat with drugs and measure their sensitivities, it will be better to proof or mention that the cells were still alive in the tissue-based ex vivo sensitivity assays.
- What is the difference between traditional staining for tumor biopsies and this tissue-based ex vivo sensitivity assays? Please add it at section Discussion or Conclusion.
Minor comments:
- Figure 2 no error bar.
- line 132: Phospho-H3 (p-H3) should provide the detailed phosphorylated site such as Ser 10?
- Figure 3, 4, 5: Please describe what are the M-series number in the figure legends.
- Please move the conclusion at section 4 to “5. Conclusions”
Author Response
We thank the reviewer for constructive comments on our manuscript. The comments were addressed as follows.
Major comments:
- The overall cisplatin sensitivity score is not clear. Please provide a detailed information for it in section of Materials and methods.
We adapted a sentence in the methods section: ‘Ex vivo sensitivity scores were determined for HE, EdU and TUNEL staining. Various cut-off values were explored and the overall cisplatin sensitivity score was calculated as the mean of the three parameters for a chosen cut-off.’ (lines 155-157)
2. Is these tissue-based ex vivo sensitivity assays only suitable for breast tumor? Please also discuss about the potential application to other cancers.
In principle this method is applicable to all types of tumors, at least if ex vivo culture conditions can be determined. We added this to the last sentence to the conclusion section (‘and can be extended to other cancer types and treatments‘).
3. line304: What are the detailed steps for embedding biopsies in agarose?
We extended the description to enable researchers to replicate our procedures (lines 112-114).
4. Do the cells alive in the tissue-based ex vivo sensitivity assays? Because the authors use tissue-based ex vivo sensitivity assays to treat with drugs and measure their sensitivities, it will be better to proof or mention that the cells were still alive in the tissue-based ex vivo sensitivity assays.
The slices are indeed alive during treatment, as is clear from the EdU incorporation data. This has been described in more detail in our previous papers about the subject (such as reference 2).
5. What is the difference between traditional staining for tumor biopsies and this tissue-based ex vivo sensitivity assays? Please add it at section Discussion or Conclusion.
This point has also been discussed in previous papers (references 2 and 13): this method captures functional parameters rather than a static biomarker. We think this issue has been discussed in sufficient detail in these previous papers.
Minor comments:
- Figure 2 no error bar.
This figure does not show a response with average and a standard deviation, but just the number of samples in our limited set that would show a significant reaction when a certain cut-off has been chosen. The only way to add error bars would be if we assume a normal distribution in the responses, which cannot be assumed a priori. Therefore, we do not think error bars are meaningful in this particular figure.
2. line 132: Phospho-H3 (p-H3) should provide the detailed phosphorylated site such as Ser 10?
The information has been added (indeed serine 10).
3. Figure 3, 4, 5: Please describe what are the M-series number in the figure legends.
The information has been added to the figure legends.
4. Please move the conclusion at section 4 to “5. Conclusions”
This has been done.
Reviewer 3 Report
Enjoyed reading this lovely manuscript. A few minor revision suggestions
- delete line 422 and line 423.
- Figure 2 missing error bars.
- Overall, the results are very qualitative, in great summary of drug responses in different PDXs. Also when presented in color coded summary figure, the visual effect is not direct. e.g. In Figure 3, I would be more comfortable to see some representative cell line figures to show case responsive, intermediate and resistant situations in the Figure 3B.
- Color-coded Figure 5B is a good try. Is there any way to simplify the subfigure to avoid using too many colors?
Author Response
We thank the reviewer for his positive words and constructive comments. We addressed these points as follows.
- delete line 422 and line 423.
We moved the last section from discussion to conclusions and deleted these sentences.
2. Figure 2 missing error bars.
This figure does not show a response with average and a standard deviation, but just the number of samples in our limited set that would show a significant reaction when a certain cut-off has been chosen. The only way to add error bars would be if we assume a normal distribution in the responses, which cannot be assumed a priori. Therefore, we do not think error bars are meaningful in this particular figure.
3. Overall, the results are very qualitative, in great summary of drug responses in different PDXs. Also when presented in color coded summary figure, the visual effect is not direct. e.g. In Figure 3, I would be more comfortable to see some representative cell line figures to show case responsive, intermediate and resistant situations in the Figure 3B.
Supplementary figure 1 presents representative figures from the PDX models for cisplatin and figure 3A shows a representative figure for taxane treatment of a primary breast tumor. The quantification of two breast tumor samples for taxane sensitivity can be found in figure 3B, where each symbol represents one field of view. The heterogeneity in the sample can be deduced from these data. Similar analysis was done for each sample and the averages were plotted in figure 3C. We think that the original pictures would not add significantly to this analysis, but if deemed necessary, they could be added as a supplementary figure.
4. Color-coded Figure 5B is a good try. Is there any way to simplify the subfigure to avoid using too many colors?
We simplified this figure by replacing color coding of biopsy site with a letter code.